

# Feasibility and benefits of group-based exercise in residential aged care adults: a pilot study for the GrACE programme

Samantha Fien[1], Timothy Henwood[1,2], Mike Climstein[3] and Justin William Leslie Keogh[1,4,5]

[1] Health Science and Medicine, Bond University, Robina, Australia
[2] School of Human Movement & Nutrition Sciences, University of Queensland, Brisbane, Queensland, Australia
[3] Exercise, Health and Performance Faculty Research Group, University of Sydney, Sydney, NSW, Australia
[4] Human Potential Centre, Auckland University of Technology, Auckland, New Zealand
[5] Cluster for Health Improvement, Faculty of Science, Health, Education and Engineering, University of the Sunshine Coast, Sippy Downs, Sunshine Coast, Australia

Corresponding author
Samantha Fien,
samantha.fien@student.bond.edu.au

## ABSTRACT

The objective of the study was to examine the feasibility and benefits of a group resistance training exercise programme for improving muscle function in institutionalised older adults. A feasibility and acceptability study was designed for a residential aged care (RAC) facility, based on the Gold Coast, Australia. Thirty-seven adults, mean age $86.8 \pm 6.1$ years (30 females) living in a RAC facility. Participants were allocated into an exercise (n = 20) or control (n = 17) group. The exercise group, the Group Aged Care Exercise (GrACE) programme, performed 12 weeks of twice weekly resistance exercises. Feasibility was measured via recruitment rate, measurement (physiological and surveys) completion rate, loss-to-follow-up, exercise session adherence, adverse events, and ratings of burden and acceptability. Muscle function was assessed using gait speed, sit-to-stand and handgrip strength assessments. All intervention participants completed pre- and post-assessments, and the exercise intervention, with 85% (n = 17) of the group attending $\geq$ 18 of the 24 sessions and 15% (n = 3) attending all sessions. Acceptability was 100% with exercise participants, and staff who had been involved with the programme strongly agreed that the participants "Benefited from the programme." There were no adverse events reported by any participants during the exercise sessions. When compared to the control group, the exercise group experienced significant improvements in gait speed ($F(4.078) = 8.265$, $p = 0.007$), sit to stand performance ($F(3.24) = 11.033$, $p = 0.002$) and handgrip strength ($F(3.697) = 26.359$, $p < 0.001$). Resistance training via the GrACE programme is feasible, safe and significantly improves gait speed, sit-to-stand performance and handgrip strength in RAC adults.

## INTRODUCTION

Ageing can lead to an impaired physical function, mobility and reduction in quality of life (*Krist, Dimeo & Keil, 2013*). A decrease in mobility may prompt a vicious cycle of

sedentary behaviours, reduced physical activity and deconditioning, with residential aged care (RAC) adults shown to be more sedentary than their community-dwelling counterparts (*Reid et al., 2013*). The mobility decline may reflect the emergence of sarcopenia, which is defined as the progressive and generalised loss of skeletal muscle mass and subsequent muscle function (muscle strength and physical performance) associated with the ageing process (*Cruz-Jentoft et al., 2010*). The preferred sarcopenic measure for physical performance in older adults is gait speed, which is also considered a primary precursor to age-related adverse events including disability, cognitive impairment, falls, mortality, institutionalisation and hospitalisation (*Abellan van Kan et al., 2009*; *Cruz-Jentoft et al., 2010*; *Peel, Kuys & Klein, 2013*). The threshold to be considered as having normal or above habitual gait speeds is 0.8 m/s (*Kuys et al., 2014*), a value almost identical to the 0.82 m/s cut-off proposed as being predictive of death within two years for older men (*Stanaway et al., 2011*).

A meta-analysis of 2,888 long-term ambulant RAC residents reported a mean habitual gait speed of 0.48 m/s (95% confidence interval (CI) 0.40–0.55) (*Kuys et al., 2014*). However, it was cautioned that the true mean gait speed of RAC adults may be even less than 0.48 m/s as many of the reviewed studies utilised non-randomly selected samples, meaning the participants were likely to be more mobile than those who did not consent to participate. Consistent with such a view, a recent study of 102 randomly selected RAC residents reported a mean gait speed of 0.37 m/s (*Keogh et al., 2015*). The widespread low gait speed documented for RAC adults and the link between low gait speed and many adverse age-related effects suggests that further research needs to be conducted to examine feasible and efficacious approaches to improving or at least offsetting the expected annual decline in gait speed of 0.03–0.05 m/s per year (*Auyeung et al., 2014*; *Onder et al., 2002*).

Two recent reviews have examined the potential for exercise, and specifically progressive resistance training (e.g. strength) and weight bearing exercise (e.g. balance and mobility) to improve many aspects of muscle function including gait speed in RAC/frail older adults (*Chou, Hwang & Wu, 2012*; *Valenzuela, 2012*). In their meta-analysis of 225 participants across four studies, *Chou, Hwang & Wu (2012)* reported that exercise produced a significant 0.07 m/s (CI 0.02–0.11) increase in gait speed compared to the control group (−6% change). However, a limitation of this literature is that the implementation of these exercise programmes in RAC is still relatively uncommon. This lack of translation may reflect the many barriers to the sustainability of resistance combined with weight bearing training programmes in RAC (*Federal Interagency Forum on Aging-Related Statistics, 2004*) and to our knowledge, a complete lack of research quantifying the feasibility of this form of exercise in this setting.

A possible exception is an exercise programme, which was targeted at respite care older adults in Australia (*Henwood, Wooding & de Souza, 2013*). Older adults accessing respite care are unable to completely care for themselves due to the adverse effects of ageing, chronic disease, physical and/or cognitive disability and are at increased risk of entry into RAC. These individuals typically access respite day care for several hours per day for one or more days per week to allow their carer the opportunity to attend to other everyday

activities or to have a break from their caregiving responsibilities. An analysis of the exercise programme demonstrated a high feasibility for translation into an ongoing respite day care centre and that 2 h of participation per week for 20 weeks significantly improved functional capacity and balance among participants (*Henwood, Wooding & de Souza, 2013*). While this programme was feasible and effective in a disabled community-dwelling population, it is yet to be trialled amongst RAC adults. Given the demonstrated uptake of this exercise programme by a low functioning older adult population at risk of entry into RAC, it was hypothesised that the Group Aged Care Exercise (GrACE) programme would exhibit similar levels of feasibility and benefits in RAC adults (*Henwood, Wooding & de Souza, 2013*).

The primary aim of this study was to determine the feasibility of the GrACE programme in RAC, with the secondary objective of measuring the programme benefits on gait speed, sit to stand and handgrip strength.

## MATERIAL AND METHODS

### Participants

Participants were included if they were:

i.    Aged 65 years and over;
ii.   Residing in a RAC;
iii.  Able to walk with a walker and/or walking stick or could self-ambulate; and
iv.   Could provide informed consent.

The exclusion criteria included:

i.    End-stage terminal and/or life expectancy < 6-months (ethical reasons);
ii.   Two person transfer or unable to self-ambulate (due to increased falls risk);
iii.  Unable to communicate or follow instructions (personal needs beyond the scope of this project);
iv.   Insufficient cognitive function to provide informed consent; and
v.    Dangerous behaviours that would endanger the client or research staff.

### Study design and recruitment

This study compared the delivery feasibility and outcomes of a 12-week combined resistance and weight bearing exercise programme, which we named the GrACE programme. Participant recruitment and assessment occurred over a five-month period. The flow of recruitment to assessment is represented in Fig. 1.

The RAC was approached about participation via email and telephone follow-up. Potential participants were identified at a meeting with the facility Service Manager. Participants were screened via the inclusion criteria at the meeting with the Service Manager and a Registered Nurse, whom also deemed who would be able to perform the exercises due to the inclusion and exclusion criteria. The Service Manager and a Registered Nurse created two lists from the eligible participants, one that contained the names of

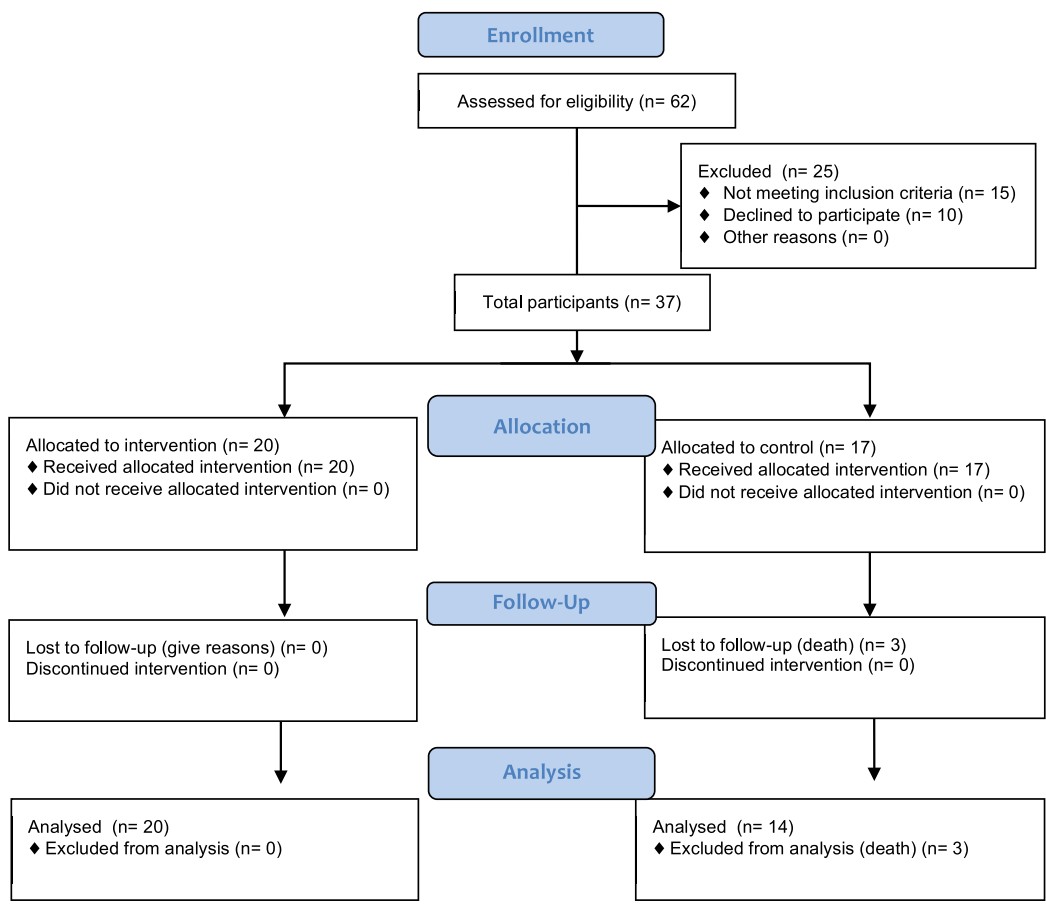

**Figure 1 Project CONSORT diagram of recruitment and assessment of study participants.** Enrolment numbers and withdrawals, or those lost to follow-up, are indicated in the boxes.

residents who could be recruited for the exercise group and one for the recruitment of the control group. This group allocation was based on the location of their bedroom with respect to the training room, as the Service Manager and a Registered Nurse felt that only participants who resided on the same level as the exercise room were likely to join and adhere to the GrACE programme. As we wished to get some idea on the number of participants who would enrol in such an exercise program, the sample obtained in the current study reflected the maximum number of participants who were eligible and provided their informed consent to participate. The final sample obtained was a convenience sample from one RAC facility. Following an explanation of the procedures, purposes, benefits and associated risks of the study, participants had the opportunity to ask questions. A total of 37 older RAC adults (86.8 ± 6.1 years, range 72–99 years, 30 females) provided written informed consent for the study. The exercise group contained 20 participants (86.9 ± 5.7 years, range 72–97 years, 15 females) and the control group 17 participants (86.3 ± 6.6 years, range 75–99 years, 15 females). Ethical approval to conduct this study was attained from Bond University's Human Ethics Research Committee (RO 1823). The protocol for this trial was published at Clinical Trial Registry ID NCT02640963.

## Intervention: the GrACE programme

Previous work by our group trialed a successful exercise programme in respite day care that could promise benefits to those in RAC (*Henwood, Wooding & de Souza, 2013*). The GrACE programme's full outline is available in Appendix 1. In brief, the programme included a number of targeted weight-bearing exercises (using body weight and dumbbells) and a range of seated, non-resisted upper- and lower-body dynamic and reaching movements. While developed for respite care older adults, the programme was slightly modified for the RAC setting; initially using reduced range of motion and resistance, and an extended conditioning/familiarisation phase. The conditioning phase lasted for three weeks in which technique was emphasised without using any weights or additional resistance. The focus of this technique of the conditioning phase was to develop the correct technique and minimise the potential for any delayed onset muscle soreness or adverse effects. After concluding the conditioning phase, participants were able to use light dumbbells (often starting with 0.5 kg) increasing to heavier dumbbells (up to 4 kg) with their increasing capacity over the course of the programme.

Participants performed the exercises twice per week for 12 weeks, with an average of 15 of the 20 participating residents attending each exercise class. Training sessions lasted approximately 45 min, were separated by at least 48 h and were delivered by an allied health professional experienced working with older adults. The sessions were conducted in the communal dining room, where the furniture was moved around prior and post training. The dining room was selected as the facility in which the exercise programme was performed had three levels, with the dining room located on the level having the highest number of residents. The allied health professional (exercise physiologist) was not blinded to the allocation of participants as they collected both pre- and post-outcomes for the study as well as conducting the exercise programme. The exercise physiologist was experienced working with community dwelling older adults, but received additional training prior to the project delivery via the respite community-training package used in a previous study (*Henwood, Wooding & de Souza, 2013*) and by RAC facility staff on issues relevant to working with RAC residents.

## Control group

All subjects assigned to the control group were given the option to engage in other activities that were offered by the facility during the 12-week intervention period. Activities were conducted either in their fitness room or communal areas, and included Zumba Gold, aerobic exercise and walking. These sessions lasted for 30–45 min and conducted by their facility's leisure staff. However, no specific resistance exercises were offered in these activities.

## Data collection

Reasons for refusal (non-consent) to participate were recorded (*Henwood et al., 2014*). All muscle function outcome measures in this study have been previously validated for use with older adults, and their protocols reported elsewhere (*Henwood, Wooding & de Souza, 2013*; *Sterke et al., 2012*). Assessments were completed one-on-one with each

participant, assessing muscle function as well as a range of demographic characteristics, which are important in describing the sample. During muscle function measures assessments, participants were encouraged to rest as needed and given verbal support and encouragement to reduce any potential burden to the participant.

# MEASURES

## Feasibility outcomes

The assessment of feasibility was defined by recruitment rate, measurement (physiological and surveys) completion rate, loss-to-follow-up, exercise session adherence, acceptability and adverse events (*Bower et al., 2014*; *Peddle-McIntyre et al., 2012*; *Suttanon et al., 2013*). Recruitment rate was defined as the number of residents recruited from those invited. Measurement completion rate was defined as the number of participants able to complete each outcome measure at baseline and follow-up. Loss to follow-up was defined as participants who withdrew or dropped out and did not consent to a follow up assessment. Exercise session adherence was measured by the number of sessions attended out of the maximum 24 sessions. Consistent with previous exercise studies involving low functioning older adults (*Bossers et al., 2014*; *Bower et al., 2014*; *Henwood, Wooding & de Souza, 2013*), the proportion of participants who completed 75 and 100% of the required 24 sessions was recorded. Acceptability was measured via a programme satisfaction survey completed post-training that assessed the burden of training and testing, as well as how participants felt about the trial. Questions included:

- "Prior to commencing the exercise programme did you have any concern(s) with the GrACE programme?";
- "Did you enjoy participating in the GrACE programme?";
- "Do you believe the GrACE programme was well organised?";
- "Whilst participating in the GrACE programme do you believe that the programme impeded on your daily routine?";
- "Would you be happy to continue participating in the GrACE programme or something similar?"; and
- "Overall, would you rate your current physical condition to be better than before you started the GrACE programme?"

Answers to the acceptability questions were scored on a five-point Likert Scale (1 = strongly agree, agree, neither, disagree or 5 = strongly disagree).

Adverse events were defined as incidents in which harm or damage resulted to a participant and included, but were not limited to, falls and fall-related injuries, musculoskeletal or cardiovascular incidents and problems with medication and medical devices (*Government of Australia, 2014*). These adverse events were recorded via the facility's records. The exercise group also received a diary to record if they had any muscle soreness or complaints about the exercise class. These diaries were returned to the instructor at the end of each week. The exercise instructor also verbally confirmed the information contained in these diaries with each of the exercise group participants at the end of each week.

## Muscle function measures outcomes

### Gait speed

Gait speed was recorded via the GaitMat II system (Manufacturer is EQInc; Model is GaitMat II), which required participants to walk across a level pressure mat system 3.66 m (11.91 ft.) long (*McDonough et al., 2001*). Participants completed the trials at their preferred (habitual) walking (gait) speed. The following instructions were given, "Walk towards the end of the room at a pace that is comfortable for you." Participants were allowed to walk in their own footwear. All measures were initiated from a standing start 2 m (6.56 ft.) from the GaitMat II platform as suggested by *Kressig & Beauchet (2004)* to reduce the effect that acceleration may have on gait speed. The average gait speed (m/s) from three attempts was used for data analysis. Participants were allowed as much rest as required between attempts, with rest periods typically being up to 1 min.

### Handgrip strength

Upper body muscle function was measured by isometric handgrip strength. When performing the handgrip strength assessments, participants were seated, instructed to keep their elbow at 90° and asked to squeeze a handgrip dynamometer (Sammons Preston Roylan, Bolingbrook, IL, USA) to their maximum ability for a period of up to 5 s (*Mathiowetz, 2002*). Three trials were performed with the subject's dominant hand with one-minute rest between trials and the best result used for analysis (*Roberts et al., 2011*).

### Sit to stand performance

The sit-to-stand measure was performed to assess lower body muscle function of the participants. In the sit-to-stand measure, participants sat and stood to their full standing position from a 43 cm high chair as many times as possible in 30 s whilst keeping their arms crossed against their chest (*Millor et al., 2013*). Timing commenced when the assessor gave the command "go."

## Participant demographics

All participants were assessed for Body Mass Index (BMI), body fat percentage (%) and cognitive status at pre- and post-testing. BMI was calculated based on body mass (kg) divided by the square of height ($m^2$). Body fat was estimated via Bioelectrical Impedance Analysis (BIA, Maltron BF-906 body fat analyser) (*Senior et al., 2015*). The BIA required participants to have two electrode stickers placed on their hand as well as two on their foot whilst in the supine position. The flow of electrical signals was measured through fat, lean tissue and water, which was then applied to a database of algorithms revealing the whole body analysis (*Chien, Huang & Wu, 2008*). Cognitive status was quantified via the Mini Cog (*Borson et al., 2000*). The Mini Cog was scored out of three words that are recalled after drawing the face of a clock on a piece of paper to read 10 min after eleven o'clock, with scores > 1–2 recalled words and abnormal clock drawing, indicative of cognitive impairment (*Borson et al., 2000*).

**Table 1 Descriptive data for the residents in the exercise control and group.**

| Variable | Exercise group (n = 20) | Control group (n = 17) |
|---|---|---|
| Age (yrs) | 86.9 ± 5.7 | 86.3 ± 6.6 |
| Range (yrs) | 72–97 | 75–99 |
| No. of females (no. %) | 15 (75%) | 15 (88%) |
| Length of stay in RAC (days) | 745.1 ± 622.6 | 755.0 ± 492.1 |
| Medical conditions (no.) | 15.3 ± 5.2 | 14.2 ± 5.3 |
| Medications (no.) | 14.3 ± 6.1 | 14.6 ± 5.9 |
| Use of walking aid | 11 (55%) | 13 (76%) |
| BMI (kg/m$^2$) | 26.5 ± 3.7 | 27.2 ± 4.7 |
| Body fat % | 33.2 ± 10.8 | 38.8 ± 5.6 |
| Marital status | | |
|    Married | 4 (20%) | 2 (12%) |
|    Divorced | 3 (15%) | 0 (0%) |
|    Widowed | 13 (65%) | 15 (88%) |
| Nationality | | |
|    European | 20 (100%) | 15 (88%) |
|    Asian | 0 | 2 (12%) |
| Primary language | | |
|    English | 19 (95%) | 16 (94%) |
|    German | 1 (5%) | 0 |
|    Russian | 0 | 1 (6%) |
| Mini-COG status # | | |
|    Positive | 12 (60%) | 14 (82%) |
|    Negative | 8 (40%) | 3 (18%) |

**Note:**
   Data are mean ± standard deviation; yrs, years; no., number.

## Statistical analysis

Descriptive statistics were calculated to describe the baseline characteristics and feasibility results, with all continuous variables presented as mean and standard deviation (±SD), and for categorical variables as the total number and percentage (%) of responses. In circumstances where participants were unable to complete a physical measure, they were given the lowest score, generally zero. All data were initially checked for normality prior to analysis by investigating homogeneity of regression slopes, scatterplots for linearity, kurtosis, skewness and Levene's test of equality of error variances. Baseline characteristics of the two groups were compared using ANCOVA and chi-square analysis for continuous and categorical variables, respectively. A one-way ANCOVA was performed to assess the between-group changes in gait speed, handgrip strength and sit to stand performance. SPSS (version 20) was used for data analysis; statistical significance was set at $p \leq 0.05$ a priori.

## RESULTS

Descriptive characteristics of the sample, provided in Table 1, showed no significant difference between groups at baseline. Of the 62 individuals put forward for participation

**Table 2 Exercise group participant questionnaire post 12-week completion.**

| Question |
|---|
| Prior to commencing the exercise programme did you have any concern(s) with the GrACE programme? |
| SA = 0    A = 0    U = 1    D = 15    SD = 4 |
| Did you have any concern(s) with the GrACE programme upon completing the exercise programme? |
| SA = 2    A = 18    U = 0    D = 0    SD = 0 |
| Did you enjoy participating in the GrACE programme? |
| SA = 20    A = 0    U = 0    D = 0    SD = 0 |
| Do you believe the GrACE programme was well organised? |
| SA = 20    A = 0    U = 0    D = 0    SD = 0 |
| Whilst participating in the GrACE programme do you believe that the programme impeded on your daily routine? |
| SA = 0    A = 0    U = 0    D = 17    SD = 3 |
| Would you be happy to continue participating in the GrACE programme or something similar? |
| SA = 20    A = 0    U = 0    D = 0    SD = 0 |
| Overall, would you rate your current physical condition to be better than before you started the GrACE programme? |
| SA = 20    A = 0    U = 0    D = 0    SD = 0 |

**Note:**
    SA, Strongly agree; A, Agree; U, Unsure; D, Disagree; SD, Strongly disagree.

**Table 3 Changes in the muscle function outcomes for the exercise and control groups.**

| Muscle function outcome | Exercise group | | | Control group | | | Between-group significance |
|---|---|---|---|---|---|---|---|
| | Pre | Post | % Change | Pre | Post | % Change | |
| Gait speed (m/s) | 0.65 ± 0.19 | 0.68 ± 0.17 | +4.6 | 0.64 ± 0.16 | 0.60 ± 0.16 | −6.0 | 0.007* |
| Sit to stand (repetitions) | 0.0 ± 0.0 | 6.4 ± 4.5 | NA | 0 ± 0 | 1.3 ± 3.2 | NA | 0.002* |
| Handgrip strength (kg) | 15.2 ± 5.3 | 15.9 ± 5.9 | +4.6 | 13.2 ± 4.5 | 10.6 ± 4.1 | −19.7 | 0.001* |

**Notes:**
    All data were reported as mean ± standard deviation; m/s, metres per second; kg, kilogram; NA, not applicable as the pre-test score equalled zero.
    * Statistical significance ($p < 0.05$).

by the RAC Service Manager, 47 were found to be eligible and 37 consented to involvement. At follow-up, three participants in the control group had passed away, resulting from falls complications (n = 1) and pre-existing heart disease (n = 2). Apart from these three deaths, the study experienced a 100% retention rate of surviving participants and final analysis was conducted on 34 individuals (20 intervention and 14 control). Seventeen (85%) participants in the exercise group attended 18 or more exercise sessions, with three (15%) of these 17 participants attending all 24 training sessions.

Participant responses to the exercise programme questionnaire, measured by the Likert scale questions are summarised in Table 2. Acceptability of the programme was very high with 100% of exercise participants and staff who had been involved with the programme (i.e. nurses who helped bring the residents from their rooms to the GrACE programme or observed the class from the nurses' station) strongly agreeing that the participants "Benefited from the GrACE programme" and "Happy to continue participating." Refer

to Appendix 2 to see the full responses to the open-ended questions. With respect to diary completion, all diaries were returned with five partially completed (ranging from 50–80%) and 15 fully completed. There were no adverse events reported by any participants during the exercise sessions.

The exercise group had significantly greater improvements in habitual gait speed ($F(4.078) = 8.265$, $p = 0.007$), sit to stand performance ($F(3.24) = 11.033$, $p = 0.002$) and handgrip strength ($F(3.697) = 26.359$, $p < 0.001$) when compared to the control group. Pre- and post-intervention muscle function measures data are presented in Table 3.

## DISCUSSION

This study demonstrated a combined weight bearing and resistance training exercise programme which we called the GrACE programme, designed and tested in community-dwelling respite day care older adults, is feasible, safe and effective in improving gait speed, sit to stand performance and handgrip strength in RAC adults. Findings from this study may assist RAC providers and care staff to develop and implement feasible, safe and effective exercise programmes for their residents.

The recruitment rates for this exercise trial appeared similar or better than other similar trials in respite care, hospital inpatient or RAC populations (Bower et al., 2014; Henwood, Wooding & de Souza, 2013; Peddle-McIntyre et al., 2012; Suttanon et al., 2013). Sixty-two of the 151 residents were identified by the service manager as being potentially eligible for the study, a proportion equating to approximately 40% of total population of the RAC facility. The other 89 RAC residents were deemed ineligible due to the following: being in a wheelchair or restricted to bed duties, not being able to follow simple instructions, lack of attention or unpredictable behaviour. It is interesting to note that some residents (n = 10) refused to participate due to fear of change in their schedule, fear of never doing resistance training before and not wanting to try and after being in the RAC facility for over five years had declared they weren't doing exercise anymore. Suggested ideas to overcome this were through word of mouth via residents to other residents, especially at communal times such as breakfast, where the residents remind each other of exercise class.

Of the 62 residents identified by the service manager as being eligible to participate, 37 (60%) provided informed consent and were placed into either the exercise or control groups. While a recruitment rate of 60% of the eligible participants may appear relatively low, this value appeared comparable (Henwood et al., 2015) or slightly higher (Álvarez-Barbosa et al., 2014; Bossers et al., 2014; Sievänen et al., 2014) than previous exercise trials in RAC. The reason provided by the 25 (40%) potentially eligible participants who declined to participate included: not being interested in the study or exercise programme, lack of time: didn't want to commit to the 12 weeks, didn't want to try something different/that was out of the normal routine and were happy with their current lifestyle. The reasons reported by the potential participants who declined participation in this project should be taken into account by future researchers considering conducting RAC exercise RCTs if they wish to ensure maximum participant recruitment and statistical power of the resulting analyses.

For the 20 participants who were enrolled into the GrACE programme, adherence rates were high with 85% of the participants attending at least 18 (75%) of the required 24 exercise classes. This relatively high attendance rate appeared similar (*Álvarez-Barbosa et al., 2014*; *Bossers et al., 2014*; *Sievänen et al., 2014*) or greater than (*Hassan et al., in press*; *Henwood et al., 2015*) previous feasibility exercise studies involving RAC older adults. Such results suggest that many exercise class options (if offered) may be well attended by residents of RAC facilities. The acceptability of the programme was further assessed by using a five-point Likert scale questionnaire that focused on the participants' perceptions of the exercise sessions. All exercise participants strongly agreed that they obtained substantial benefits from their participation and that they wished to continue being involved in the programme. These results are consistent with previous studies reporting high acceptability of RAC exercise participation (*Álvarez-Barbosa et al., 2014*; *Bossers et al., 2014*; *Henwood et al., 2015*; *Sievänen et al., 2014*). However, in contrast, our study supports the feasibility of an in-centre delivery using targeted supervision and inexpensive equipment, where previous work has employed more complicated deliveries. Specifically, *Bossers et al. (2014)* delivered individualised, supervised walking and strength training programmes five days per week, *Sievänen et al. (2014)* and *Álvarez-Barbosa et al. (2014)* delivered their interventions using relatively expensive whole body vibration devices, *Hassan et al. (in press)* used expensive resistance training machines and *Henwood et al. (2015)* trialled aquatic exercise that required participants to be transported to and from a community pool and change in and out of swimming attire. Our study required only one qualified trainer; the exercises were conducted at the RAC facility so there was no transport needed and delivery was not affected by busy times or school holidays such as experienced by *Henwood et al. (2015)*. Further, our study didn't need to have a personalised exercise programme developed for each individual resident or require the purchase and storage of expensive equipment (*Álvarez-Barbosa et al., 2014*; *Hassan et al., in press*; *Sievänen et al., 2014*). The acceptability of the programme was further demonstrated by the lack of any adverse effects reported within the exercise group. This lack of adverse effects is again consistent with the literature on a variety of exercise programmes for RAC residents (*Álvarez-Barbosa et al., 2014*; *Bossers et al., 2014*; *Hassan et al., in press*; *Henwood et al., 2015*; *Sievänen et al., 2014*). Therefore, our study further supports the safety of supervised exercise in this population, and demonstrates that the perception held by some care staff that exercise is a dangerous for RAC residents is not based on the current peer-reviewed evidence.

However, it must be acknowledged that although the GrACE programme was found to be feasible for those who participated in this study, this amounted to only ~25% of the population of the RAC facility. Collectively, the results of this study suggest that further feasibility trials may need to target RAC residents who were ineligible for this study (*Gibbs et al., 2015*) and also examine some of the issues influencing recruitment rates from those who were eligible to participate (*Kalinowski et al., 2012*).

While the primary focus of this study was to demonstrate feasibility and acceptability of the GrACE programme in RAC, we were also interested in further quantifying the benefits of exercise in this population as such data may inform future RCTs in this area. Significant between-group differences were reported for gait speed, sit to stand performance and

handgrip strength, all of which favoured the exercise over the control group. Such results are impressive due to the relative simplicity of the GrACE programme performed in the current study and the importance of these outcome measures for older adults, particularly those living in RAC who wish to maintain their health and independence.

The significant between group effect for gait speed was a promising finding, with the relative 0.07 m/s improvement in gait speed for the exercise group was identical to the results of a recent meta-analysis involving exercise for RAC residents (*Chou, Hwang & Wu, 2012*). However, results of the within-group analysis indicated that the exercise group experienced an increase in gait speed of 0.03 m/s (~5%), whereas the control group experienced a decline of 0.04 m/s (~6%). As the control group's decline of 0.04 m/s was consistent with the expected annual decline of 0.03–0.05 m/s per year for older adults (*Auyeung et al., 2014*; *Onder et al., 2002*), the GrACE programme appears able to maintain or perhaps slightly increase gait speed in this population.

The significantly greater improvement in sit-to-stand performance for the exercise group was of considerable interest, with none of the residents being able to perform the sit-to-stand with their hands on the chest during baseline assessments. The ability to safely rise from a chair and sit down is a prerequisite for maintaining Activities of Daily Living (ADL) function (*Zijlstra et al., 2012*). Specifically, older adults who are unable to perform the sit to stand are considered below the capacity for independence and at risk of accelerated physical deterioration associated to extended sedentary behaviours (*Zijlstra et al., 2012*). The RAC participants' lack of leg strength (as demonstrated by their initial inability to perform even one sit to stand) was consistent with previous research that observed over 70% of RAC adults were unable to rise from a chair without assistance (*Sabol et al., 2011*).

The exercise group's significantly greater improvement in handgrip strength was also considered important, as lower handgrip strengths have predicted an accelerated decline in ADL disability and cognition as well as functional limitation and physical disability in older adults (*Bohannon et al., 2007*; *Hairi et al., 2010*; *Taekema et al., 2010*). The exercise-related increase in handgrip strength was also consistent with a previous exercise study in RAC (*Justine et al., 2012*).

## Study limitations

There were several limitations in the current study. We acknowledge that 37 participants assessed represented a recruitment rate of only 59.6% of the participants initially thought to be eligible for the study and only about 25% of the facility's population. However, this rate of uptake is not uncommon in RAC, nor is the strict eligibility criteria imposed or participant safety reasons in exercise intervention. Therefore, caution is warranted if considering this GrACE programme feasibility across all RAC populations. It should also be noted that there was a risk of bias with the same allied health practitioner conducting the outcome measures and supervising the exercise programme. Future RCT should address this limitation by blinding the assessor to the participants' group allocation. The significant improvements observed in the present study for gait speed, sit to stand performance and handgrip strength also need to be replicated in RCTs due to the lack of a controlled study design and attention matching of the participants of each group.

## CONCLUSION

The GrACE programme, consisting of progressive weight bearing combined with resistance training, was shown to be feasible, safe and effective in improving muscle function in RAC residents. Improved muscle function measurements are valuable client outcomes, and may have significant cost saving benefits to the RAC setting, as increased muscle function could reduce the RAC residents' degree of disability, care needs and risk of falls (*Chou, Hwang & Wu, 2012*). This work has direct measurable benefits for RAC residents, staff and providers and other health professionals working with older adults. Our findings addressed a number of previously unanswered and understudied questions in relation to the feasibility, safety and benefits of exercise classes that could be offered by RAC facilities to their residents. By having a greater understanding of these issues, RAC providers can better target interventions (e.g. exercise for maintaining gait speed or to reduce falls risk) to their residents. We would therefore encourage other RAC providers to strongly consider implementing similar programmes.

### Clinical messages

- Progressive resistance and weight bearing training, as performed in the GrACE programme, is feasible and safe for RAC residents.
- Progressive resistance and weight bearing training may significantly improve gait speed, sit-to-stand performance and handgrip strength in RAC residents.

## ACKNOWLEDGEMENTS

We would like to thank the management, staff and residents of the RAC for their assistance and participation in this project.

### Funding

The authors received no funding for this work.

### Competing Interests

Justin William Leslie Keogh is an Academic Editor for PeerJ.

### Author Contributions

- Samantha Fien conceived and designed the experiments, performed the experiments, analyzed the data, contributed reagents/materials/analysis tools, wrote the paper, prepared figures and/or tables, reviewed drafts of the paper.
- Justin William Leslie Keogh conceived and designed the experiments, analyzed the data, contributed reagents/materials/analysis tools, wrote the paper, reviewed drafts of the paper.
- Timothy Henwood conceived and designed the experiments, reviewed drafts of the paper, developed exercise programme that GrACE was developed from.
- Mike Climstein conceived and designed the experiments, reviewed drafts of the paper.

## Clinical Trial Ethics

The following information was supplied relating to ethical approvals (i.e., approving body and any reference numbers):

Bond University's Human Ethics Research Committee (RO 1823).

## Data Deposition

The SPSS data sheets are supplied as Supplemental Files.

## Clinical Trial Registration

The following information was supplied regarding Clinical Trial registration:

ClinicalTrials.gov Identifier: NCT02640963

## Supplemental Information

Supplemental information for this article can be found online at http://dx.doi.org/10.7717/peerj.2018#supplemental-information.

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
