# Peer review of "Feasibility and benefits of group-based exercise in residential aged care adults: a pilot study for the GrACE programme"

_PeerJ, doi:10.7717/peerj.2018_

## Round 0.1 · original submission · Major Revisions

The manuscript addresses clearly an important clinical question.
It will certainly take some effort to address the comments and concerns of the reviewers, in particular the concerns of reviewer #1, but I am confident that it will be worth the effort. It may also be helpful for the reader to know how the analysis plan for the study was developed and when it was executed.

·

Basic reporting

Well presented Feasibility study.

Experimental design

The study focusses on Feasibility and Acceptability. I would suggest focussing on this as there was an emphasis on efficacy, which would not be so appropriate at this stage (see general feedback).

Validity of the findings

Feasibility data well presented. Would like to see greater discussion about why there was a low uptake and the implication of this for the main trial.

Additional comments

Thank you for the opportunity to review the manuscript from Dr Fein and colleagues. The paper reports the feasibility and efficacy of a group based exercise programme in residential aged care adults.

Strengths:
- Important clinical question
- Well presented
- Clear statement of feasibility

Weaknesses
- Focus on functional outcome measures

General comments.

This is a well written and thought through paper. The focus of the paper is on Feasibility and Acceptability and I would encourage the authors to emphasise the latter and play down the functional changes (as a Feasibility and Acceptability study it is not deisgned to look at efficacy but lay the foundation for a larger RCT). I hope that my reflections assist the authors in improving their paper.

- The emphasis should be on feasibility and acceptability (although the latter was only measured in the treatment group). The discussion talked at length about the functional benefits of the programme and comparison to other studies, yet the study was not really designed to do this. I would suggest that the authors consider discussing more the low uptake rate and how this could be addressed in the main study, and de-emphasise the efficacy measures.

- In terms of the functional outcomes, there is clearly a strong effect. However, the study is not designed to look at efficacy of the intervention, so please discuss the changes in light of power and how they would inform the main trial (power / sample size etc.).

- It would be worth discussing that the comparison group were not attention matched. As wellbeing measures are affected by contact, this is worth discussing as a way that you could improve the main trial.

- Figure 1 should show the number of potential recruits.

- Looking at the Trial registration, the trial was not prospectively registered (first registered Dec 15 2015). If the trial was an RCT, this could be problematic. But, if the study were a feasibility and acceptability (and not efficacy), this would not be a problem.

·

Basic reporting

This article appears to meet all the basic reporting requirements but there were some editing issues that should be addressed, such as:
- Some of the citations throughout the article do not have spaces between the author's names. All citation should be checked to ensure this is rectified.
- The first time the term 'GrACE' is used in the article it is not written in full (line 71) but instead written in full after the second time it is used (lines 73-74). The complete term for GrACE should be written in full followed by '(GrACE)' at the first use (line 71) and then the abbreviation used after that point.
- The opening sentence in the second paragraph of the results section (starts on line 216) could be structured better so that the outcome measure being discussed is clear, as currently when it is read initially it is not clear what measure the Likert scale is related to.
- Table 3 needs to column widths altered so that the words 'significance' (column 8) and 'change' (columns 4 and 7) are written on a single line rather then across two.
- The sentence on lines 276-278 could be constructed differently so it is clearer.
- The sentence that concludes on line 331 with ".......current study due to." appears to be incomplete, this needs to be checked and rectified.
- There should not be a capital 'P' for progressive on line 344.

Experimental design

A more descriptive method would be beneficial to ensure there is sufficient detail to be reproducible. Examples include providing further detail on:
- The randomisation process: how it was done?; who completed it and were they independent from the study and/ or blinded?
-How was the sample size determined?
- The allied health professional that completed the outcome measures and activity: was there more than one?; was it the same person that completed both tasks?; what type of allied health professional?; were they independent of the study?; was the allied health professional that completed the outcome measures blinded to group allocation if possible?; did they receive any extra training to complete their role in the study?
- The control group: further detail about the type of activities they were involved in as part of their 'usual care', including how often, how long, type of setting and who it was conducted by would be beneficial.
- Intervention group: how many participants were in each exercise session at a time?;what was the environment/setting like, e.g. in a communal area, dedicated fitness room etc?
- Sit to stand test: Further detail such as height of chair(s) used, what it aims to measure and at what point does the timing start, e.g. when assessor says "go" or when the participant starts to attempt to stand.
- Handgrip strength: In this section it states "upper body function was assessed using isometric hand grip strength and sit-to stand performance" (lines 185-86). I believe this might just be a typographical error as the sit-to-stand t4est helps to determine lower limb function, particularly when upper limbs are not used to complete the test, as was the case in this article.
- Outcome measures: It may be beneficial to report further detail on the validity and reliability of the outcome measures rather then just stating thy have been found to be valid and reliable in previous studies (line 131-2).

Validity of the findings

In the conclusion there were just a couple of minor suggestions:
- It is stated "improved muscle function measurements may have significant cost saving benefits to the RAC setting" (lines 346-347), it would be beneficial to give a couple of examples to support this statement.
- In the 'clinical messages' bullet points at the end of the article it would be a good idea to include a point about the feasibility of the exercise program as the this was the primary focus in the research question.

Reviewer 3 ·

Basic reporting

This manuscript meets all the requirements as set by PeerJ policies.
There are some issues with punctuation and formatting that need to be addressed.
• Several references appear in an odd format, no spaces or commas between authors.
• Line 92 – insert comma after ‘programme’
• Line 127 – insert semi-colon after ‘walking’ and comma after ‘however’
• Line 155 – should this read, “Did you find the GrACE programme impeded (not impede) on your daily routine?”
• Line 200 – data is a plural term. Use ‘were’ instead of ‘was’
• Line 276 – insert comma after ‘however’
• Line 277 – delete the word ‘with’
• Line 331 – sentence …”current study due to…” Due to what?
• Line 344 – ‘progressive’ does not need a capital

Experimental design

There is no description of how body fat was obtained, yet this is reported in Table 1. Similarly, the details of height and weight for BMI also need to be included as does the Mini-cog test. How were participants randomised?

Validity of the findings

Some discussion as to the clinical significance of physical performance changes would be useful. The significant difference between the groups is obviously affected by the large decline in the controls rather than any great improvement in the exercising group. The gait speed in both groups was below the cited threshold, some further insight into any clinical significance pre- to post-intervention in both groups would be useful. The increase in gait speed in the exercise group is minimal. I am not sure this demonstrates the ‘effectiveness’ of the programme. Would the authors like to proffer some discussion as to the death of three participants, all from the control group? There is no discussion on the health professional’s reflection of delivering the programme

---

## Round 0.2 · Minor Revisions

Please try to follow the reviewer's recommendation and check for overall consistency and flow, taking into account that the original focus had to shift in a somewhat different direction.

·

Basic reporting

This article appears to meet all the basic reporting requirements and all comments raised in my previous review have been addressed.

Experimental design

All the comments raised in the previous review have been addressed.

I note that the authors has advised that true randomisation was not used by the individuals that completed group allocation so this term has been removed from the study. However, on reviewing the revised manuscript 'random allocation' was still mentioned in the abstract and exercise recruitment consort diagram, they need to be reviewed and updated. The CONSORT checklist also needs to be revised now that 'random allocation' has been removed from the study.

The authors has clarified information about the AHP who conducted the outcome measures and intervention as suggested. Because the same AHP conducted the outcome measures and intervention the authors may want to consider adding this to the limitation section due to the risk of bias associated with this fact.

Validity of the findings

This article appears to meet all the validity of findings requirements and all comments raised in my previous review have been addressed.

Reviewer 3 ·

Basic reporting

The authors have addressed the majority of the reviewers comments to an adequate standard. However, some further attention to punctuation is needed. In addition, three paragraphs in the Discussion, beginning on lines 401, 411, 423 all begin with "The significant between-group effect for...". The repetition is unnecessary and needs some thought to keep the reader engaged. This is the Discussion, not the Results section.

Experimental design

This study was originally designed as an RCT, which was not achieved in the end. Reviewer 2 asked for determination of sample size. Surely, if this was an RCT, a sample size calculation would have been undertaken? The final sample is a convenience sample and should be acknowledged as such.

I agree with Reviewer 3, the question "Did you find the GrACE program impede on your daily routine?" is awful; the tense is not correct.

Validity of the findings

No comments

Additional comments

I have some issues with the 'fluidity' of this work. The purpose seems to have shifted in response to the aspects highlighted by the reviewers. It moved from feasibility and benefits (with efficacy a key) to feasibility and acceptance. However, despite significant limitations, I think it has interest.

---

## Round 0.3 · accepted · Accept

Thank you for taking the time and energy to deal with the various comments and suggestions.

·

Basic reporting

This article appears to meet all the basic reporting requirements and all comments raised in my previous review have been addressed.

Experimental design

This article appears to meet all the experimental design requirements and all comments raised in my previous review have been addressed.

Validity of the findings

This article appears to meet all the validity of the findings requirements and all comments raised in my previous review have been addressed.

Additional comments

Nil